# Investigation of the Effects of Multi-Wall and Single-Wall Carbon Nanotubes Concentration on the Properties of ABS Nanocomposites

**Brenda Janett Alonso Gutierrez** [1,2], **Sithiprumnea Dul** [3,*], **Alessandro Pegoretti** [3], **Jaime Alvarez-Quintana** [1] **and Luca Fambri** [3,*]

1   Centro de Investigación en, Materiales Avanzados S. C. Unidad Monterrey, Alianza Norte # 202, Autopista Monterrey-Aeropuerto Km.10., Apodaca C.P. 66600, Nuevo León, Mexico; brenda.alonso@cimav.edu.mx (B.J.A.G.); jaime.alvarez@cimav.edu.mx (J.A.-Q.)
2   Facultad de Ingeniería Mecánica y Eléctrica, Universidad Autónoma de Nuevo León, Avenida Universidad s/n, Ciudad Universitaria, San Nicolás de los Garza C.P. 66451, Nuevo León, Mexico
3   Department of Industrial Engineering and INSTM Research Unit, University of Trento, 38123 Trento, Italy; alessandro.pegoretti@unitn.it
*   Correspondence: sithiprumnea.dul@unitn.it (S.D.); luca.fambri@unitn.it (L.F.); Tel.: +39-0461-283728 (S.D.); +39-0461-282413 (L.F.)

**Abstract:** The effects of two types of carbon nanotubes, namely multiwall (MWCNT) and single-wall (SWCNT) carbon nanotube, on the thermal and mechanical properties of acrylonitrile-butadiene-styrene (ABS) nanocomposites, have been investigated. ABS filled-CNT nanocomposites with various filler loadings of 5–10 wt% were properly produced by a solvent-free process in blend compounding at 190 °C. Compression moulded plates and extruded filaments were obtained at 190 °C and 230 °C, respectively. Melt flow index (MFI), shore hardness, Vicat temperature, differential scanning calorimeter (DSC) and thermogravimetric analysis (TGA) were performed to characterize and compared the different CNT nanocomposites. ABS/SWCNT composite filaments showed higher tensile properties (i.e., stiffness and strength), than ABS/MWCNT. The electrical resistivity of ABS/SWCNT and ABS/MWCNT filaments decreased to 0.19 Ω.cm and 0.65 Ω.cm for nanocomposites with 10 wt% of nanofillers; a power law was presented to describe the electrical resistivity of composites as a function of the CNTs content. A final comparative parameter regarding melt flow, stiffness and conductivity was also evaluated for understanding the combined effects of the nanofillers. SWCNT nanocomposites exhibited better overall cumulative results than MWCNT nanocomposites.

**Keywords:** carbon nanotubes; nanocomposites; ABS; thermal properties; mechanical properties; solvent-free compounding; conductivity

## 1. Introduction

In the past few decades, polymer nanocomposites (PNCs) have been a great expansion in research to fabricate multifunctional materials with remarkable properties, such as high mechanical, thermal and electrical performance [1,2]. The development of polymer nanocomposites can lead to light-weight structural materials with functionalities that can be utilized in broad applications such as in electronic components [3,4], micro-batteries [5], circuits [6] and electromagnetic shielding [7]. Polymer nanocomposites are referred to the polymeric material (i.e., thermoplastic, thermoset, or elastomer) typically consisting of one or more nanoscale materials (nanoparticles). The advantages of the utilization of polymers are due to their cost effectiveness, reproducibility, less suffering from corrosion and easy manufacturing processes [8].

Among these nanoparticles, a different form of carbon nanotubes and graphene have been commonly utilized due to their extraordinary intrinsic properties [9–12]. In particular, carbon nanotubes (CNTs) is 2D nanomaterial which is rolled sheets of a hexagonal array

of carbon atoms. CNTs were first observed by Baker et al. [13] in the 1970s but were re-discovered by Iijima et al. in 1991 [14]. Carbon nanotubes are defined by the number of concentric walls: single-walled carbon nanotubes (SWCNT) having only a single layer of graphitic carbon atoms, and multi-walled carbon nanotubes (MWCNT) with several layers of coaxial carbon tubes. Several synthesis methods of carbon nanotubes have been used to reach a variety of diameters, aspect ratio, crystallinity, crystalline orientation, purity, and surface chemistry. In particular, CNTs are generally produced by three main techniques: electric arc discharge, laser ablation, and chemical vapour deposition (CVD) [15]. In addition, CNTs have been widely investigated as nanofillers due to their remarkable physical, mechanical and electrical properties. Carbon nanotube is known as an excellent nanofiller for increasing the electrical conductivity of nanocomposites, even at lower concentration (<1 wt%) [9,16].

The dispersion level of nanoparticles in the polymer matrix can be improved by the treatment of their surface to aggregation phenomena within the nanocomposites. Due to very small size of the colloid particle, the overall surface forces at the interface of the particles are large, resulting in agglomerates. In order to separate and prevent particles's agglomeration, the particles must hold the sample electrical charge (positive or negative) to produce a force of mutual electrostatic repulsion [17]. In previous researches, various surface treatment methods were applied in order to increase the content of functional group of nanoparticles, including $O_2$ plasma, nitric acid, nitric acid/sulfuric acid, acid/oxidizer, ozone/heat, UV/ozone, and amine grafting treatment [18,19]. The investigation of the interfacial properties is important for the understanding of the structure/properties relationships governing the mechanical behaviour of polymer nanocomposites [20]. Using a compatibilizer within the polymer matrix and functional nanoparticles could enhance the filler/matrix interface. Organosilane compounds, maleic-anhydride-grafted (MA-g) organic fatty acid derivatives, and MA-grafted petroleum-based polymers (e.g., MA-g high-density polyethylene (HDPE), MA-g polypropylene (PP)) are commonly used as compatibilizers and/or coupling agents. For examples, the possible chemical esterification reactions may occur between -COOH groups of compatibilizers and the -OH groups on the surface of oxidized carbon fibre [21].

Carbon-based nanofillers, e.g., graphene nanoplatelets (GNP) and carbon nanotubes, were successfully incorporated in various matrices such thermoplastic ABS [9,16], in crosslinked matrixes, such as epoxy [22–24], in degradable polymers (e.g., polylactic acid (PLA)) [25–27], or even in thermoplastic polyurethanes (TPUs) [28]. The positive effect of nanofillers was confirmed by the improvement of various physical properties in particular mechanical properties and electrical conductivity. In view of specific applications and processing, high-performance nanocomposite of poly-ether-ether-ketone (PEEK) and CNT were also properly compounded for the production of electrically conductive filaments, suitable for fused filament fabrication (FFF), also known as 3D printing [29].

Our previous works have been focused on the preparation of acrylonitrile butadiene styrene (ABS) nanocomposites containing carbonaceous nanofillers such as graphene, carbon nanotube and carbon black by melt-compounding [30–33]. Specific attention was also paid to the preparation of nanocomposite filaments (ABS) for 3D printing containing MWCNT and/or graphene in order to obtain products with high mechanical properties and low electrical resistivity by using the FFF technique [30,33]. In particular, it was shown that the addition of MWCNT improved highly the electrical conductivity, but significantly increased the viscosity of materials that consequently required high-temperature FFF process (280 °C). We also observed that proper mixing of 3 wt% MWCNT and 3 wt% GNP provide a possible compromise between relatively easy processability and an acceptable enhancement of performances (i.e., mechanical and specific electrical properties), resulting in the production of ABS nanocomposite filaments for 3D printing with resistivity of 4.1 Ω.cm [34]. We also found that the electrical percolation threshold of nanocomposites was achieved at 0.9 wt% for MWCNT [16]. In our previous work [33], CNT nanocomposite samples exhibited an acceptable and a good electrical conductivity at 4 wt% and 6 wt%

of MWCNT, respectively. There is a general agreement in literature that the resistivity decreases with the filler concentration, so the authors decided to explore the preparation of nanocomposites with CNT equal and higher than 5 wt% in order to be suitable for thermoelectric applications.

Hence, in this work, we focus on the incorporation of MWCNT or SWCNT nanofillers up to 10 wt% into the ABS matrix by a solvent-free process (melt compounding); and on the preparation of samples by compression moulding and filament extrusion. The microstructures and properties (i.e., physical and electrical properties) of the produced nanocomposites are compared to highlight the potential offered by each nanofiller.

## 2. Materials and Methods

### 2.1. Materials

Multiwall (MWCNT) and single wall (SWCNT) carbon nanotubes used in this work were obtained from commercial producers, with the trade name MWCNT-NC7000 (Nanocyl CA, Sambreville, Belgium) [35], and SWCNT-TUBALL (Tuball$^{TM}$, Columbus, OH, USA) [36]. Details of carbon purity, and selected physical properties are presented in Table 1.

**Table 1.** Properties of commercial multi-walled and single-walled carbon nanotubes according to the manufacturer.

| Nanotubes | Carbon Purity (%) | Density (g/cm$^3$) | Length (μm) | Diameter (nm) | Aspect Ratio | Surface Area (m$^2$/g) | Manufacturer |
|---|---|---|---|---|---|---|---|
| MWCNT-NC7000 | >90 | 2.15 ± 0.03 * | 1.5 | 9.5 | 158 | 250–300 | Nanocyl, Belgium |
| SWCNT-TUBALL | >80 | 1.877 ** | 5 | 1.6 ± 0.4 | 313 | >300 | TUBALL, USA |

\* Data from Reference [33]; ** data from Reference [36].

Acrylonitrile butadiene styrene (ABS) with tradename Sinkral® F322 was provided by Versalis S.p.A. (Mantova, Italy). According to the manufacturer's technical data sheet, the material has a density of 1.04 g/cm$^3$ and melt flow rate of 14 cm$^3$/10 min (220 °C/ 10 kg) [37].

All materials were dried at 80 °C in a vacuum oven for at least 2 h before processing.

### 2.2. Material Processing and Sample Preparation

#### 2.2.1. Compounding

MWCNT and SWCNT nanofillers at a concentration of 5, 7.5 and 10 wt% were mixed with ABS by melt compounding in Thermo-Haake Polylab Rheomix counter-rotating internal mixer (Thermo Haake, Karlsruhe, Germany) operated at a temperature of 190 °C and a rotor speed of 90 rpm for 15 min. After the mixing, all composites were grinded, by a granulator (Piovan, model RN 166, S. Maria di Sala (VE), Italy), into small-sized particles for further processing.

#### 2.2.2. Compression Moulding (CM)

Compounded materials were hot-pressed in a Carver Laboratory press (Carver, Inc., Wabash, IN, USA) at a temperature of 190 °C under a load of 10 tonnes applied for 10 min to obtain rectangular plaques with dimensions 160 × 80 × 4 mm.

#### 2.2.3. Filament Extrusion

The compounded materials were frozen in liquid nitrogen and furtherly milled into fine powders by using a grinder IKA M20 Universal mill (IKA®-Werke GmbH&Co. KG, Staufen, Germany). Powder materials after sieving under a strainer pore size of 1.0 mm were also used to feed a single-screw extruder model Estru 13 (Friul Filiere SpA, Buia-UD, Italy), with a screw diameter of 14 mm and rod die diameter of 2 mm. The temperature profile of the extruder gradually increased from $T_1$ = 100 °C to $T_2$ = 210 °C, $T_3$ = 220 °C,

and $T_4$ = 230 °C (rod die). The screw rotation speed was fixed at 30 rpm, and collection rate was regulated by using a take-up unit Thermo Electron to achieve a final diameter of the filament of 1.75 ± 0.05 mm (see Figure 1).

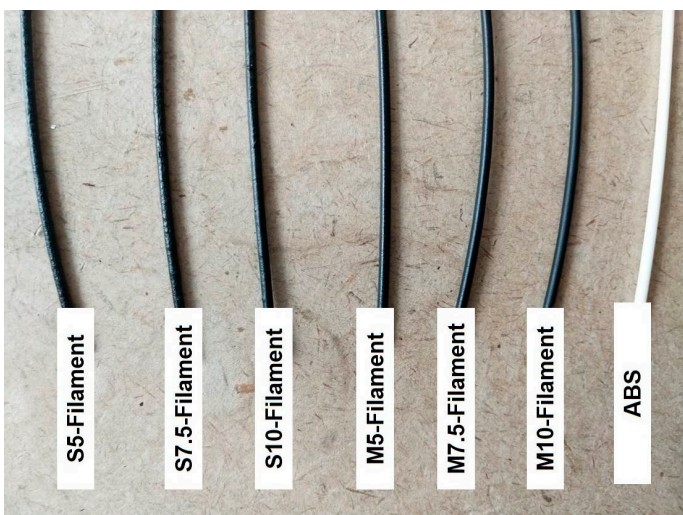

**Figure 1.** Extruded filaments of neat acrylonitrile butadiene styrene (ABS), and ABS/SWCNT and ABS/MWCNT nanocomposites (S and M filaments, respectively) at 5 wt%, 7.5 wt%, and 10 wt% of nanofiller (filament code of nanocomposites indicates the type of CNT and the percentage).

*2.3. Testing Techniques*

### 2.3.1. Scanning Electron Microscopy (SEM)

Nanocomposite CM samples and filaments were broken in liquid nitrogen, and their fracture surfaces were observed by a Carl Zeiss AG Supra 40 Field-Emission Scanning Electron Microscope (FESEM) (Carl Zeiss AG, Oberkochen, Germany). Representative micrographs were taken at an acceleration voltage of 6 kV and at different magnification (100x and 50Kx).

### 2.3.2. Density Measurement

Density measurements of the bulk composite compression molded samples and filaments was performed according to the standard ASTM D792-13 [38]. Moreover, the theoretical density and the voids content in nanocomposites were evaluated through the rule of mixture, as detailed in our previous work [33]. The results are the average of five measurements.

### 2.3.3. Shore Hardness Test

The Shore D hardness was measured following ASTM D2240-05 [39] on CM specimens that were 4 mm thick by using an ATS-Faar S.p.A durometer (Milano, Italy) at 23 °C, where the average of 10 measurements in different positions after an indentation time of 3 s was used for analysis.

### 2.3.4. Vicat Softening Temperature (VST)

The Vicat softening temperature was evaluated using an HDT-VICAT instrument (ATS-Faar S.p.A., Milano, Italy) according to ASTM D1525-09 [40]. Three CM specimens, each with a 4 mm thickness, were tested for their compositions at a heating rate of 120 °C/h under an applied load of 10 N.

### 2.3.5. Melt Flow Index (MFI)

The MFI test was carried out at 280 °C and 10 kg following to the ASTM D 1238-10 [41] standard (procedure A) by using a Kayeness Co. model 4003DE capillary rheometer

(Morgantown, PA, USA). Filament samples about 5 g were cut and tested after a pre-heating and compaction time of about 5 min. The results are the average of five measurements.

### 2.3.6. Thermogravimetric Analysis (TGA)

Thermal degradation analysis was performed using a Q5000 IR thermogravimetric analyzer (TA Instruments–Waters LLC, New Castle, DE, USA). Each sample had a mass of about 10 mg and was tested in a nitrogen flow of 15 mL/min up to 700 °C at a heating rate of 10 °C/min. The onset temperature of the degradation ($T_{onset}$) was determined by the intersection point of the two tangent lines, and the maximum temperature ($T_{d,max}$) was defined by the maximum of the first derivative of the weight loss. The residual weight at selected temperatures was also measured.

### 2.3.7. Differential Scanning Calorimetry (DSC)

DSC analyses of filaments were investigated by a Mettler DSC 30 calorimeter (Mettler Toledo, Greifensee, Switzerland). Samples have a mass of about 10 mg and were tested under heating-cooling-heating cycle in the range between 30 °C and 260 °C at a constant rate of $\pm10$ °C/min at a nitrogen flow of 100 mL/min. Glass transition temperature (Tg) was determined at the inflection point of thermogram, and the variation of specific heat at transition ($\Delta C_P$) was evaluated in the first and second heating step.

### 2.3.8. Mechanical Test

Uniaxial tensile test on filaments was carried out at a crosshead speed of 10 mm/min by using an Instron® 5969 electromechanical tester (Norwood, MA, USA) with a load cell of 50 kN. Elastic modulus of filaments (diameter of 1.75 mm; total length of 150 mm; gauge length of 100 mm) was determined as a secant value between strain levels of 0.05% and 0.25%; yield and fracture properties were also measured, as an average value of five replicates.

### 2.3.9. Electrical Resistivity Measurement

The test was performed on composite filaments at room temperature, by using a four-probes contact configuration, according to ASTM D4496–04 [42] standard for moderately conductive materials. Samples were applied under a voltage of 5 V by a DC power supply (IPS303DD produced by ISO-TECH, Milan, Italy). The current flow between external electrodes and voltage between internal electrodes on the samples was recorded through an ISO-TECH IDM 67 Pocket Multimeter electrometer (ISO-TECH, Milan, Italy). Five specimens per each sample were tested, and the electrical volume resistivity of the samples was determined according Equation (1) as follows:

$$\rho = R \cdot \frac{A}{L} \tag{1}$$

where $R$ is the volume resistance, $A$ is the cross-section of the specimen (diameter of 1.75 mm), and $L$ is the length between the internal electrodes ($L = 3.69$ mm).

## 3. Results and Discussion

Preliminar evaluation and comparison has been performed on compression moulded specimens (CM), whereas in the following the extruded filaments have been deeply characterized in view of possible applications where polymeric materials with low electrical resistivity are required

### 3.1. Compression Moulding

Bulk nanocomposites were tested in terms of structure, and morphology, and defects. Moreover, specific hardness and maximum using temperature were comparatively discussed.

3.1.1. Density and Morphology

The bulk density of both ABS/CNT compression moulded samples is reported in Table 2. In comparison with the experimental density of pure ABS matrix (1.043 g/cm$^3$), which is consistent with the reported value in the technical datasheet [36]. Density of both ABS/CNT nanocomposites increases almost linearly with the filler content up to 1.083 g/cm$^3$ and 1.086 g/cm$^3$ at 10 wt% of MWCNT and SWCNT, respectively. The calculated volume content of filler in MWCNT nanocomposites is lower than that in SWCNT nanocomposites, due to the different bulk density of CNT, being 2.15 and 1.88 g/cm$^3$ respectively (Table 1) In addition, the experimental density of ABS-filled SWCNT nanocomposites is slightly lower, compared to the theoretical density determined by using the rule of mixture. Moreover, it is evident the presence of microvoids, whose volume fraction ($V_{VCM}$) is lower than 1% as reported in Table 2, but no clear dependence on the content or on the type of filler could be observed.

**Table 2.** Composition of compression moulded samples of ABS and its nanocomposites, and correspondent experimental density, calculated volume percentage of CNT, theoretical density and void fraction ($V_{VCM}$). Hardness Shore D and Vicat Softening Temperature (VST) are also reported.

| Samples | Experimental Density (g/cm$^3$) | CNT Vol (%) | Theoretical Density (g/cm$^3$) | $V_{VCM}$ (%) | Shore D (Hs) | VST (°C) |
|---|---|---|---|---|---|---|
| ABS-CM | 1.043 ± 0.001 | 0 | 1.043 | 0 | 78.8 ± 0.4 | 111.5 ± 0.7 |
| M5-CM | 1.069 ± 0.001 | 2.49 | 1.071 | 0.2% | 79.3 ± 0.4 | 113.4 ± 1.0 |
| M7.5-CM | 1.081 ± 0.001 | 3.78 | 1.085 | 0.4% | 79.6 ± 0.7 | 115.7 ± 0.8 |
| M10-CM | 1.096 ± 0.001 | 5.11 | 1.100 | 0.3% | 80.0 ± 0.4 | 116.4 ± 0.5 |
| S5-CM | 1.066 ± 0.001 | 2.84 | 1.067 | 0.1% | 79.8 ± 0.8 | 121.3 ± 2.1 |
| S7.5-CM | 1.070 ± 0.002 | 4.31 | 1.079 | 0.8% | 78.4 ± 0.7 | 124.5 ± 2.6 |
| S10-CM | 1.086 ± 0.001 | 5.82 | 1.092 | 0.5% | 80.5 ± 0.5 | 132.2 ± 0.6 |

The fracture surfaces of compression moulded samples were analyzed by scanning electron microscopy in order to evaluate the dispersion of CNT and to evidence any defects. The most representative images of ABS/MWCNT and ABS/SWCNT at 10 wt% of filler are documented in Figure 2a–d, respectively. Fracture surface of solid CM samples at low magnification evidenced in Figure 2a,c some minor defects (microvoids, line of junction), especially in SWCNT nanocomposite, that could be attributed to the relative low temperature of processing, i.e., 190 °C. At higher magnification, MWCNT and SWCNT were clearly identified and observed in the SEM micrographs, as shown in Figure 2b,d, with evidence of uniform distribution and good dispersion. It is also worth noting the presence of long microfilament of SWCNT randomly oriented in ABS matrix (Figure 2d), as a conductive network at relatively large frame. On the other hand, a denser network could be also observed in MWCNT nanocomposite (Figure 2c).

3.1.2. Shore Hardness and Maximum Using Temperature (VST).

In order to compare the effect of different CNT on bulk properties of nanocomposites technical measurements were performed. In particular, shore hardness (Hs) and Vicat softening temperature (VST) gave interesting information of compression moulded plates related to the tip penetration at room temperature and at high temperature (near polymer $T_g$), respectively. Similarly, Shore D values of polymer nanocomposites (about 79–81 Hs) with respect to 79 Hs for ABS, regardless of the effect of nanofiller, as reported in Table 2. This experimental evidence could be explained by the dominant rigidity of the polymer over CNT at ambient temperature. On the other hand, Vicat Softening Temperature (VST) provides useful data on the maximum using temperature, because VST is depending on the materials rigidity at high temperature (see thermograms reported in Figure S1). VST value progressively increased with CNT content in the polymer matrix, as reported in Table 2. In particular, VST for nanocomposites containing 10 wt% of MWCNT was only slightly

increased from about 112 °C to 116 °C (i.e., about 4%), whereas in the case of SWCNT nanocomposites a very significant high increase was observed, reaching with 10 wt% of filler a maximum value of 132 °C (i.e., 18%).

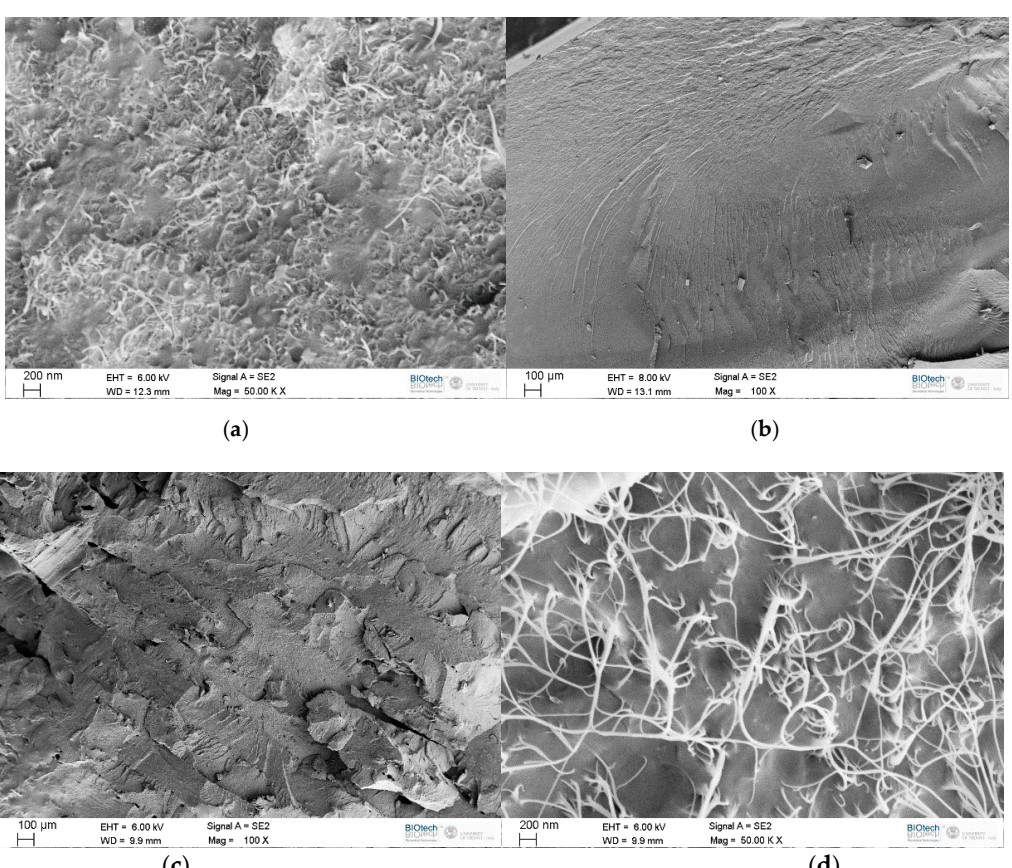

**Figure 2.** FESEM micrographs of compression moulded samples: M10-CM (**a**,**b**) and S10-CM (**c**,**d**).

*3.2. Filament Extrusion and Properties*

All the compositions were easily extruded in filament form by using a lab-scale extruder (screw with diameter of 14 mm and a compression ratio of 2.5). The main differences between the extrusion and compression-molding process were the particle sizes in the feed (milled powder instead of ground particles) and the maximum processing temperature, i.e., 230 °C instead of 190 °C.

3.2.1. Morphology and Density

The fracture surfaces of ABS/MWCNT and ABS/SWCNT filaments at 5 and 10 wt% are documented by in Figures 3 and 4, respectively. Low magnification SEM micrographs, Figure 3a,c, evidenced an almost regular fracture surface of MWCNT nanocomposite having a diameter of 1.85 mm and 1.82 mm, respectively. At higher magnification an uniform distribution and a good dispersion were clearly documented in Figure 3b,d, even with some microcavity, independently on the MWCNT content. On the other hand, ABS filled SWCNT filaments evidenced in both fracture surface of 5 wt% and 10 wt% (full cross-section of 1.70 mm and 1.83 mm as shown in Figure 4a,c) the presence of micro voids between 20 and 80 micron. In conformity to compression moulded samples, at high magnification long microfilaments of SWCNT randomly oriented were also observed in Figure 4b,d. The presence of SWCNT variously distributed on the fracture surface suggest a negligible effect of orientation and alignment in the direction of filament extrusion. Due to high aspect ratio of SWCNT, a conductive network at relatively large frame can easily formed even at relative large frame.

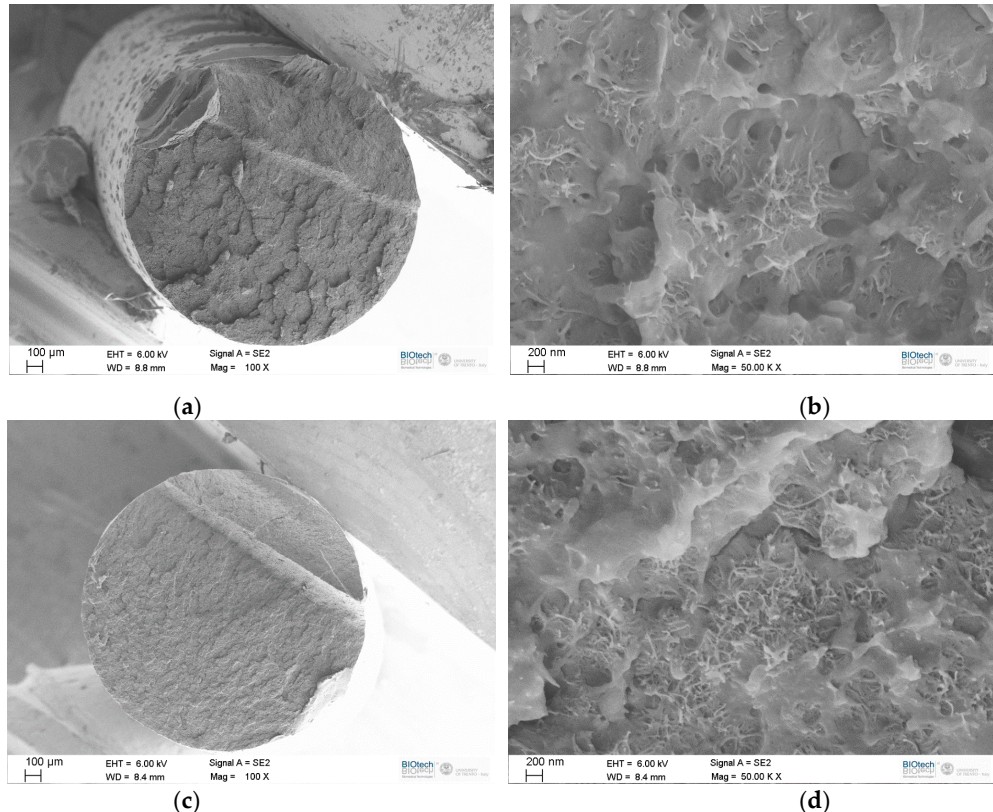

**Figure 3.** FESEM micrographs of ABS/MWCNT composite filaments: M5-Filament (**a**,**b**) and M10-Filament (**c**,**d**).

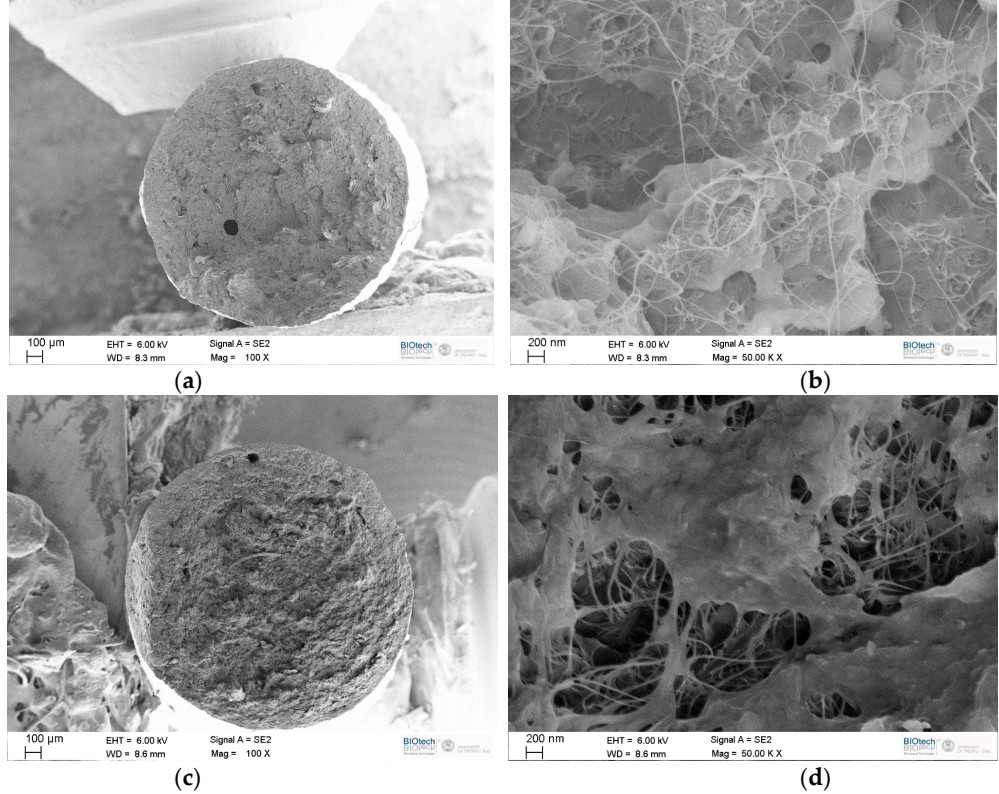

**Figure 4.** FESEM micrographs of ABS/SWCNT composite filaments: S5-Filament (**a**,**b**) and S10-Filament (**c**,**d**).

ABS/MWCNT filaments had a visually smooth lateral surface, while the rough surface on ABS/SWCNT filaments was observed. The bulk density of both ABS/CNT filaments, which is reported in Table 3. In particular, experimental density values of ABS/MWCNT and ABS/SWCNT composites were compared to theoretical density, and the void fraction ($V_{\mathrm{VF}}$) was also determined.

**Table 3.** Composition of filaments of ABS and its nanocomposites, and correspondent experimental density, calculated volume percentage of CNT, theoretical density and void fraction ($V_{\mathrm{VF}}$).

| Samples | Experimental Density (g/cm$^3$) | CNT Volume (%) | Theoretical Density (g/cm$^3$) | $V_{\mathrm{VF}}$ (%) |
|---|---|---|---|---|
| ABS-Filament | 1.041 ± 0.005 | 0 | 1.041 | 0 |
| M5-Filament | 1.069 ± 0.001 | 2.48 | 1.069 | 0.0% |
| M7.5-Filament | 1.073 ± 0.001 | 3.78 | 1.083 | 1.0% |
| M10-Filament | 1.094 ± 0.004 | 5.10 | 1.098 | 0.4% |
| S5-Filament | 1.060 ± 0.004 | 2.84 | 1.065 | 0.4% |
| S7.5-Filament | 1.070 ± 0.003 | 4.30 | 1.077 | 0.7% |
| S10-Filament | 1.076 ± 0.007 | 5.80 | 1.090 | 1.3% |

Density of both ABS/CNT nanocomposites increases up to 1.094 g/cm$^3$ and 1.076 g/cm$^3$, almost linearly with increasing the CNT content at 10 wt% of MWCNT and SWCNT, respectively.

In addition, the experimental density of ABS-filled SWCNT nanocomposites was found lower than that of MWCNT nanocomposites, as possible presence of microvoids. Table 3 shows an almost linear dependence of void fraction ($V_{\mathrm{VF}}$) with SWCNT content. Calculated volume percentage of CNT is almost similar to that of CM samples.

### 3.2.2. Thermal Properties

All DSC thermograms of ABS-CTN filaments were used for the determination of the glass transition temperature $T_{\mathrm{g}}$ either in heating or in cooling, as reported in Table 4. The first heating and second heating scan are reported in Figure S2 The $T_{\mathrm{g}}$ value associated to SAN phase in ABS/CNT filaments was determined at about 108 °C and 109 °C in the first and the second heating run, respectively, a little bit more than $T_{\mathrm{g}}$ of ABS. This means that the presence of both MWCNT and SWCNT exhibit only minor effects on $T_{\mathrm{g}}$ of ABS/CNT composites.

**Table 4.** Glass transition temperatures ($T_{\mathrm{g}}$) and different heat capacity ($\Delta C_{\mathrm{P}}$) of styrene-acrylonitrile phase in ABS and in nanocomposite (inflection point of DSC thermogram). Comparison of first and second heating up to 260 °C.

| Samples | First Heating | | Cooling | Second Heating | | $T_{\mathrm{g}}$ Comparison $T_{\mathrm{g2}} - T_{\mathrm{g1}}$ |
|---|---|---|---|---|---|---|
| | $T_{\mathrm{g1}}$ (°C) | $\Delta C_{\mathrm{P1}}$ (J/g.K) | $T_{\mathrm{gC}}$ (°C) | $T_{\mathrm{g2}}$ (°C) | $\Delta C_{\mathrm{P2}}$ (J/g.K) | $\Delta T_{\mathrm{g}}$ (°C) |
| ABS-Filament | 106.1 | 0.38 | 102.1 | 108.0 | 0.39 | +1.9 |
| M5-Filament | 108.4 | 0.34 | 99.0 | 109.7 | 0.35 | +1.3 |
| M7.5-Filament | 108.6 | 0.33 | 98.7 | 109.5 | 0.32 | +0.9 |
| M10-Filament | 108.5 | 0.25 | 98.1 | 110.9 | 0.32 | +2.4 |
| S5-Filament | 109.1 | 0.31 | 104.5 | 109.9 | 0.33 | +0.8 |
| S7.5-Filament | 108.9 | 0.27 | 101.3 | 110.0 | 0.36 | +1.1 |
| S10-Filament | 108.2 | 0.30 | 99.3 | 109.4 | 0.34 | +1.2 |

Moreover, the higher the CNT content, the lower the $T_{\mathrm{g}}$ in cooling. This phenomenon could be attributed to the presence of CNT that induced a delay in cooling transition, and a dependence of the polymer chain mobility from the rubbery to the glassy state and of the

relative free volume variation. SWCNT evidenced a more pronounced effect on the glass transition temperature in the cooling scan.

It is also interesting to note the systematic increase of $T_g$ of about 1–2 °C in the second heating scan after controlled cooling from 260 °C. In addition, the reduction of $\Delta C_P$ (variation of heat capacity) of nanocomposites with various CNT content is a direct indication of the amount of amorphous phase. In the second scan from about 0.39 J/gK (ABS) to 0.36–0.32 J/gK (CNT nanocomposites) with a tendency of higher reduction for SWCNT.

The results of DSC analysis gave interesting information not only on the materials, as compounded and produced filaments, but also on their processing in view of sequential heating and cooling in the possible steps of filament transformation. In fact typical processes of ABS nanocomposites range between 250–280 °C. These findings revealed that both MWCNT and SWCNT nanocomposites could be treated and processed at least up to 260 °C.

Thermal stability and degradation of ABS nanocomposite filaments was investigated by using thermal gravimetric analysis (TGA). Figure 5a,b depicts the TGA thermogram of ABS/CNT-filled composite filaments, while the most important parameters are summarized in Table 5. Mass loss at 280 °C was determined in order to shed more light on the behavior of processing conditions.

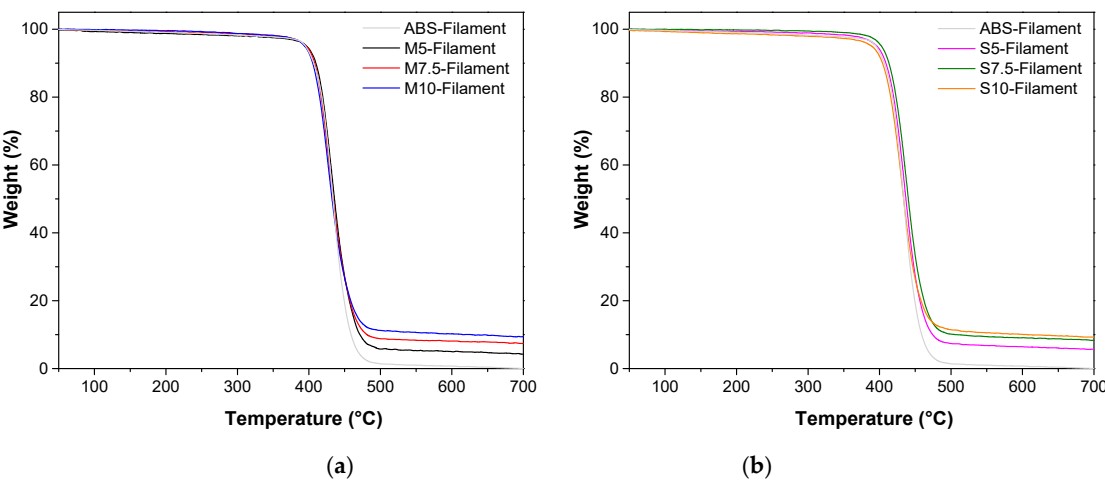

**Figure 5.** TGA curves of nanofilled ABS/MWCNT (**a**) and ABS/SWCNT (**b**) filaments compared to ABS filament under nitrogen atmosphere.

**Table 5.** TGA data of pure ABS nanocomposite filaments in a nitrogen atmosphere.

| Samples | Mass Loss at 280 °C (%) | $T_{onset}$ (°C) | $T_{d,max}$ (°C) | Residue at Selected Temperature (wt%) | | |
|---|---|---|---|---|---|---|
| | | | | 475 °C | 575 °C | 700 °C |
| ABS-Filament | 1.6 | 398.2 | 433.5 | 3.3 | 0.8 | 0.0 |
| M5-Filament | 1.8 | 399.4 | 432.2 | 8.8 | 5.2 | 4.2 |
| M7.5-Filament | 1.1 | 394.8 | 428.7 | 11.2 | 8.2 | 7.4 |
| M10-Filament | 1.0 | 382.3 | 427.3 | 13.2 | 10.5 | 9.4 |
| S5-Filament | 1.0 | 377.7 | 435.8 | 9.8 | 6.6 | 5.7 |
| S7.5-Filament | 0.5 | 384.0 | 438.0 | 13.3 | 9.2 | 8.4 |
| S10-Filament | 1.9 | 372.6 | 435.0 | 13.5 | 10.3 | 9.2 |

ABS/CNT nanocomposites decompose in one single step of degradation in an inert (nitrogen) atmosphere. This behaviour is probably attributed to the butadiene particle content in ABS structure. In addition, the relative char residue of ABS composite in Table 5 increases with the amount of nanofiller up to 9.2–9.4% at 700 °C, which is approximate to initial filler content. As reported in Table 5, the onset temperature ($T_{onset}$) and maximum

degradation temperature ($T_{d,max}$) seems to decrease with increasing CNT content. In addition, $T_{d,max}$ of ABS/MWCNT composites is slightly higher than that of ABS/SWCNT composites. This behaviour could be induced by the different thermal degradation of carbon nanotubes and its interaction with the thermoplastic. SWCNT has a higher aspect ratio and surface area than MWCNT, as documented in Figures 3 and 4.

### 3.2.3. Melt Flow Index

The processability and viscosity of the filament materials were examined by comparing their melt flow index values, as previously compared for graphene and CNT ABS nanocomposites [20]. This aspect could particularly important in view of processing, as in the case of 3D printing [12,32]. Figure 6 shows the effects on the MFI value of the nanofiller amounts and the types of CNT in nanocomposite filaments (all data are reported in Table S1). At 280 °C and 10 kg) standard ABS exhibited a melt flow value of 232 g/10 min. As expected, the MFI values of nanocomposite filaments significantly decreased up to 13–20 g/10 min with 5 wt% of nanofiller. At higher CNT content, MWCNT accounts for the much higher reduction, reaching MFI values in the range of 0.2–0.7 g/10 min. On the other hand, it is worthwhile to note that a relative low reduction up to about 2 g/10 min was observed for 10% SWCNT. The significant decrease of MFI values with the CNT content is attributed to the increasing viscosity induced by the formation of a nanofiller network.

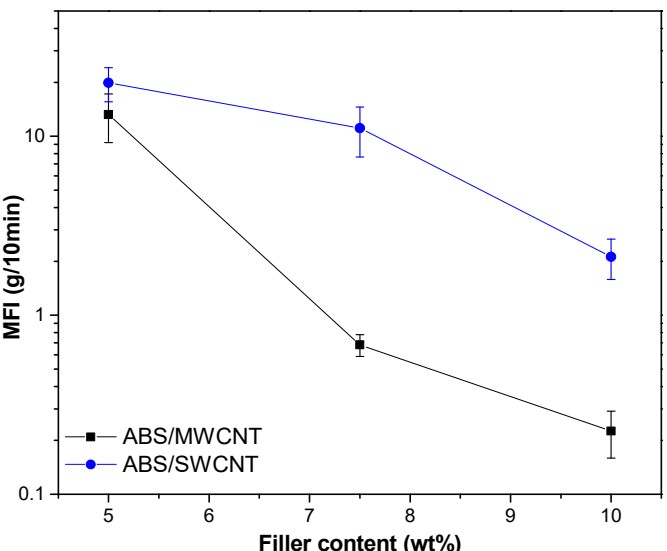

**Figure 6.** Melt flow index (at 280 °C/10 kg) of ABS/MWCNT and ABS/SWCNT nanocomposite filaments.

### 3.2.4. Tensile Properties of Filaments

Reinforcement effect of MWCNT and SWCNT in ABS nanocomposites is investigated on mechanical properties of filaments samples. Stress-strain curves are reported in Figure 7, and the tensile properties of pure ABS, ABS/MWCNT and ABS/SWCNT nanocomposites are summarized in Table 6. As expected, both ABS/MWCNT and ABS/SWCNT show an enhancement of tensile properties with CNT content. In Table 6, the elastic modulus of SWCNT-based nanocomposites is higher than that of MWCNT-based composites. For example, the elastic modulus of nanocomposite containing 10 wt% of SWCNT was increased from 2207 MPa to 6190 MPa (i.e., about 280%) whereas in the case of 10 wt% of MWCNT a corresponding value is only 2771 MPa (i.e., about 26%). The elastic modulus of the nanocomposites is influenced by the nanofiller properties such as their stiffness, shape and orientation and their dispersion level.

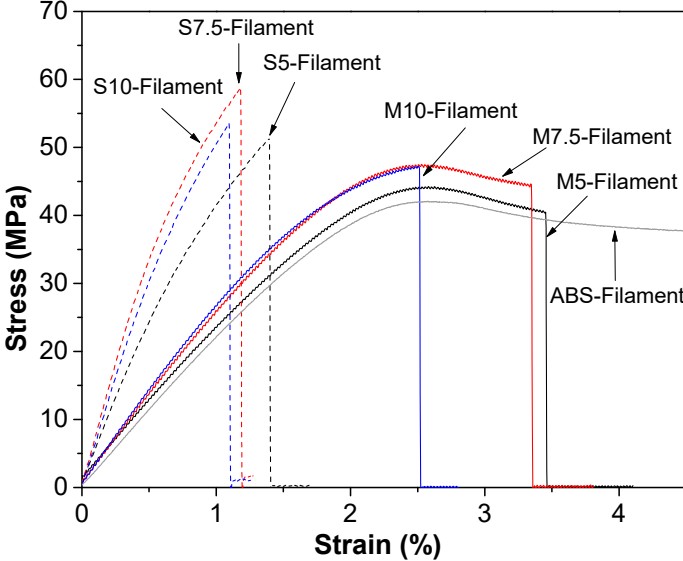

**Figure 7.** Representative stress-strain curves of ABS filament and nanocomposites filaments containing MWCNT (line) and SWCNT (dot line) up to 10 wt%.

**Table 6.** Comparison of the tensile properties of ABS/SWCNT and ABS/MWCNT nanocomposites of filaments and 3D-printed samples.

| Samples | Tensile Modulus E (MPa) | Yield Stress $\sigma_y$ (MPa) | Yield Strain $\varepsilon_y$ (%) | Stress at Break $\sigma_b$ (MPa) | Elongation at Break $\varepsilon_b$ (%) | Tensile Energy at Break TEB (MJ mm$^{-3}$) |
|---|---|---|---|---|---|---|
| ABS-Filament * | 2207 ± 65 | 42.8 ± 1.9 | 2.6 ± 0.1 | 35.0 ± 0.4 | 25.6 ± 15.8 | 8.94 ± 5.61 |
| M5-Filament | 2438 ± 142 | 44.0 ± 1.3 | 2.5 ± 0.2 | 40.2 ± 1.6 | 3.5 ± 1.3 | 1.10 ± 0.39 |
| M7.5-Filament | 2602 ± 218 | 46.9 ± 0.8 | 2.5 ± 0.2 | 44.5 ± 1.4 | 3.3 ± 0.8 | 1.06 ± 0.35 |
| M10-Filament | 2771 ± 155 | - | - | 47.5 ± 1.8 | 2.6 ± 0.2 | 0.84 ± 0.12 |
| S5-Filament | 5204 ± 275 | - | - | 52.1 ± 3.3 | 1.4 ± 0.2 | 0.44 ± 0.10 |
| S7.5-Filament | 6751 ± 506 | - | - | 53.9 ± 9.8 | 1.1 ± 0.3 | 0.37 ± 0.18 |
| S10-Filament | 6190 ± 362 | - | - | 54.6 ± 6.1 | 1.2 ± 0.2 | 0.37 ± 0.13 |

* Data from Reference [33].

In general, several factors which could affect the tensile strength of ABS nanocomposites are included the filler/matrix interfacial adhesion, its aspect ratio and dispersion level in the matrix. Table 6, shows that strength of ABS/SWCNT composite filaments is higher than that of ABS/MWCNT nanocomposites which can be attributed to the higher aspect ratio and surface area of SWCNT with ABS matrix with respect to MWCNT, as shown in Figure 4b,d. Moreover, some bending and twisting in the structure of the SWCNT could increase the bending stiffness of carbon nanomaterials with reduced dimension (and can be easier realized in single wall carbon nanotubes that in multiwall ones) [43] and prevent the detachment of SWCNT from ABS matrix. Therefore, these factors may induce a better interfacial interaction between the SWCNT and ABS matrix due to more efficiency in the load transfer from the ABS matrix to SWCNT. Consequently, tensile strength is higher for SWCNT nanocomposites. Another reason could also be considered, the 2-phase effect of 2D carbon nanomaterials stretching which also could contribute to the stretching of the nanotube containing composite [44]. that was observed in carbon nanotubes

Similarly, the strain at break was observed to be more severely reduced in the case of SWCNT nanocomposites. Tensile energy to break (TEB) directly depends on both rigidity/strength and elongation at break, and multiwall CNT nanocomposites exhibited higher values than single wall CNT nanocomposites.

### 3.2.5. Electrical Resistivity

CNT is known as an excellent filler for enhancing the electrical properties of composites. Improvements in electrical and thermal conductivity properties by adding MWCNT up to 8 wt% have been documented in our previous work [33], in particular the volume resistivity of the composites significantly decreases with at least 4 wt% of CNT.

In this case, both, MWCNT/ABS and SWCNT/ABS evidenced a significant decrease of the electrical resistivity starting from the amount of 5%wt to 10%wt of CNT content, which is several orders of magnitude lower than ABS ($10^{15}$ $\Omega$.cm), reaching about 0.65 $\Omega$.cm for M10-Filament and 0.19 $\Omega$.cm for S10-Filament. Specifically, SWCNT/ABS composites with 10%wt CNTs content exhibited three-fold better conductivity than the counterpart with MWCNT/ABS, as documented in Table 7. The volume resistivity of all samples was found directly dependent on the CNT content, and various models could be presented. It is well known that well dispersed carbon nanotubes in a dielectric matrix tend to form percolation chains which improve significantly the electrical conductivity of the system [45]. A simple power law describes the relation between composite conductivity and conductive filler concentration:

$$\rho_c = \rho_0 \left( X_f - X_{th} \right)^{-t} \tag{2}$$

where $\rho_c$, $\rho_0$, $X$, $X_{cr}$ stand for the composite resistivity, constant parameter, fraction of the filler content, and fraction percolation threshold, respectively. Likewise, parameter t is the critical exponent. Evidently, both composites follow a power-law characteristic of the electrical percolation behavior, as shown in Figure 8. From the best fit of the data to $\rho_c \left( X_f \right)$ power law, the value of *t* was evaluated; the MWCNT/ABS composites present a *t* ~ 1.69, and the SWCNT/ABS composites a *t* ~ 1.96, as shown in Figure 8a,b respectively. Previously calculated values of parameter *t* in percolation model systems of conductive fillers in an insulating matrix report values from 1.6 to 2 [46]. Besides, recently a value of *t* ~ 1.6 for CNT based composites has been reported [47], in agreement with our findings confirming the existence of the percolation mechanism in the charge transfer of our MWCNT and SWCNT composites. Hence, the interconnecting junctions of carbon nanotubes effectively act as a conducting path in percolating clusters estending across the ABS matrix.

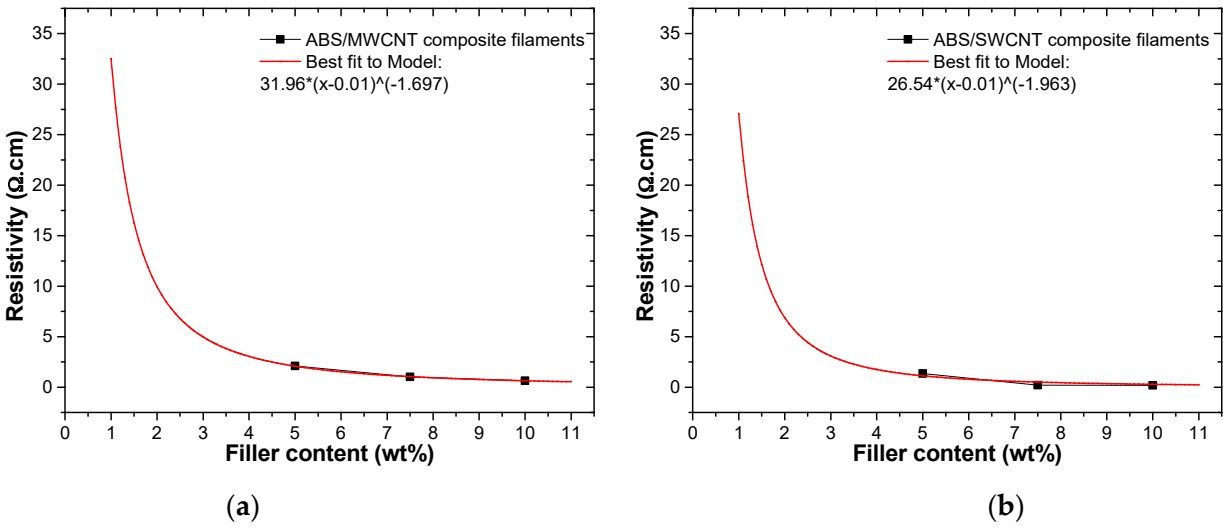

**Figure 8.** Variation of resistivity measured by experiment. (**a**) Resistivity of the MWCNT/ABS composites as a function of MWCNT wt%. (**b**) The variation of resistivity as a function of SWCNT wt%. Squared solid symbols represent experimental data, whereas red solid lines are the best fit to the percolation model.

**Table 7.** Volume electrical resistivity of ABS nanocomposite filaments.

| Samples | Resistivity (Ω.cm) |
|---|---|
| M5-Filament | 2.10 ± 0.12 |
| M7.5-Filament | 1.02 ± 0.04 |
| M10-Filament | 0.65 ± 0.04 |
| S5-Filament | 1.36 ± 0.10 |
| S7.5-Filament | 0.21 ± 0.02 |
| S10-Filament | 0.19 ± 0.03 |

*3.3. Comparative Effects of MWCNT and SWCNT*

The comparative effects of MWCNT and SWCNT as nanofillers in ABS nanocomposites and their side effects on the properties, could be evidenced in the radar plot, shown in Figure 9. Conductivity, MFI, VST and mechanical properties were selected as significant and comparable properties. The larger the area, the better the overall behavior. Figure 9 suggests that MWCNT determined a slight increase of the electrical conductivity and a quite high reduction of MFI, that could be critical for FFF processing; at the same time an almost negligible increase of Vicat, and interesting values of elongation at break were observed.

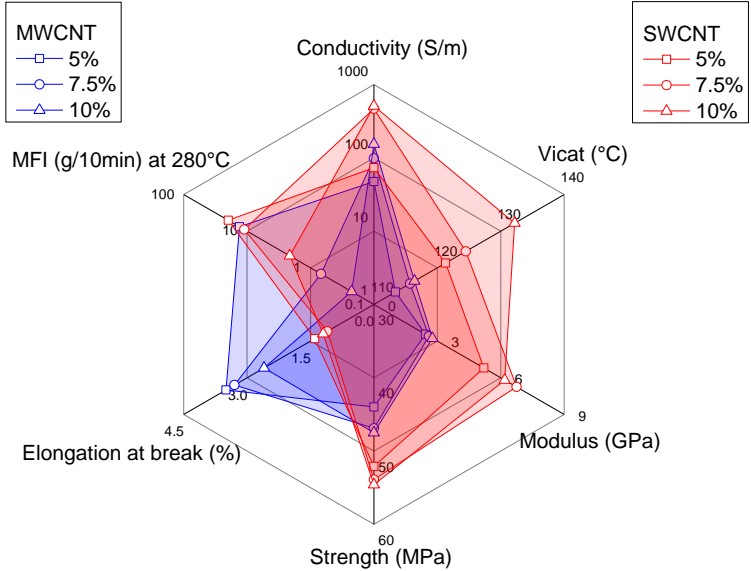

**Figure 9.** Comparison of selected properties of ABS/MWCNT and ABS/SWCNT nanocomposites as a function of nanofiller content (5–10 wt%).

On the other hand, SWCNT exhibited the higher electrical conductivity and the higher Vicat temperature; moreover, SWCNT nanocomposites revealed positive reinforcement effects on mechanical properties, such as the increase in tensile modulus and strength, but a consistent reduction of strain at break. Meanwhile, slightly lower reductions of MFI with SWCNT content.

Hence, it is possible to conclude that both MWCNT and SWCNT have beneficial and negative effects regarding the processing and the properties of these ABS nanocomposites. The quantitative evaluation of the nanofiller effects can be used for the definition of comparative factors which are associated to selected specific properties for the applications, as previously proposed for the comparison of graphene and CNT nanocomposites [16]. In particular, the processability, the stiffness and the conductivity of nanocomposites should be taken into account for defining an interesting comparative parameter $P_{M,E,S}$ according to Equation (3)

$$P_{M,E,S} = MFI \times E \times S \tag{3}$$

where *MFI*, *E* and *S* represent the melt flow index, modulus and conductivity the nanocomposites, respectively. Figure 10 shows that 5%, 7.5% and 10% of MWCNT lead to a cumulative variation of the combined factors (i.e., stiffness, processability and conductivity) that progressively decrease with CNT content.

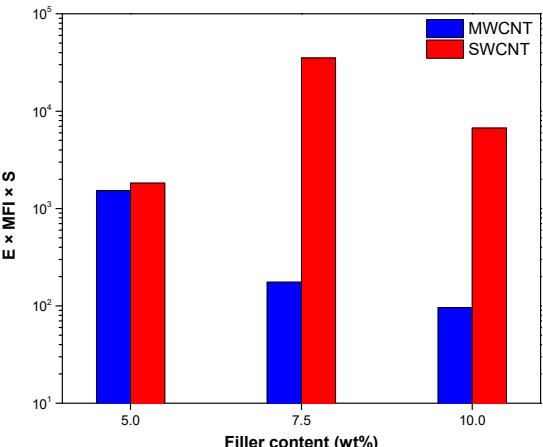

**Figure 10.** Comparison of parameters $P_{M,E,S}$ encompassing the effects of melt flow index, elastic modulus (at 280 °C and 10 kg) and conductivity, as function of nanofiller fraction up to 5–10 wt% of both MWCNT and SWCNT (See Equation (3)).

On the other hand, comparing composition at 7.5 wt% and 10 wt%, the parameter $P_{M,E,S}$ for SWCNT composites is much higher, showing in particular values 2–3 times higher than correspondent MWCNT nanocomposites. In previous work, the comparative values of MFI × E × S exhibited an optimal composition for ABS nanocomposites at 4 wt% of MWCNT [16]; in this research, a relative maximum for ABS nanocomposites at 7.5 wt% of SWCNT was determined. Interestingly, SWCNTs represent not only a valuable and useful nanofiller for improving electrical conductivity (i.e., reduction of resistivity), but also for maintaining adequate stiffness and processability.

## 4. Conclusions

Through a solvent-free mixing process, multiwalled and single-walled CNT nanofillers were properly dispersed in an ABS matrix up to concentrations of 10 wt%. The experimental study compared the effects of both nanofillers on the properties of ABS nanocomposites. The nanofillers caused an increase in modulus and tensile strength, as well as a significant reduction in the strain at break. The ABS/SWCNT nanocomposites showed a slightly higher stiffness and creep stability than the ABS/MWCNT nanocomposites. However, the tensile strength of the ABS/SWCNT nanocomposite filaments was significantly enhanced, probably due to their better orientation and stronger interfacial interactions between the SWCNTs and the ABS matrix. In addition, a more consistent reduction in MFI was observed with an increasing fraction of MWCNTs rather than SWCNTs. A significantly low resistivity in the range of 0.2–2.1 Ω.cm was obtained after the dispersion of 5–10 wt% of CNTs. In particular, ABS/SWCNT nanocomposites with 7.5 wt% and 10 wt% contents showed electrical resistivity that was 3–5 times lower than the corresponding ABS/MWCNT compositions.

These achievements can be used as a guide in the fabrication of highly conducting and reinforced CNT/ABS filaments to produce electrical and electronic parts by using Fused Filament Fabrication processes. In particular, future research could explore the potential applications of the developed 3D printable ABS/CNT composites, which can offer acceptable mechanical properties and specific electrical properties for applications such as thermoelectric devices.

**Supplementary Materials:** The following are available online at https://www.mdpi.com/article/10.3390/c7020033/s1, Figure S1: Representative Vicat softening temperature curves of acrylonitrile butadiene styrene (ABS) and nanocomposites at different content of single-walled and multiwalled carbon nanotubes (SWCNTs and MWCNTs, respectively) between 5 and 10 wt%. CM: compression-molded samples. Figure S2: Representative differential scanning calorimetry (DSC) thermograms of ABS/MWCNT and ABS/SWCNT nanocomposite filaments at different filler content between 5 and 10 wt%. First (a) and second (b) heating scans. Table S1: Melt flow index (280 °C, 10 kg) of neat ABS, ABS/MWCNT, and ABS/SWCNT nanocomposites at 5, 7.5, and 10 wt% of filler content.

**Author Contributions:** All authors conceptualized and designed the experiments; B.J.A.G. and S.D. performed the experiments; all authors analyzed the data and wrote the paper. All authors have read and agreed to the published version of the manuscript.

**Funding:** This research did not receive any external funding.

**Acknowledgments:** The authors warmly acknowledge Claudia Gavazza for technical assistance in SEM analysis.

**Conflicts of Interest:** The authors declare no conflict of interest.

## Abbreviations

The following abbreviations are used in this manuscript

| | |
|---|---|
| ABS | Acrylonitrile Butadiene Styrene |
| ASTM | American Society for Testing and Materials |
| CNT | Carbon Nanotubes |
| CM | Compression Moulding |
| DSC | Differential Scanning Calorimetry |
| FFF | Fused Filament Fabrication |
| MFI | Melt Flow Index |
| MWCNT | Multi-Wall Carbon Nanotubes |
| SEM | Scanning Electron Microscopy |
| SWCNT | Single-Wall Carbon Nanotubes |
| $T_g$ | Glass transition temperature |
| TGA | Thermogravimetric Analysis |
| VST | Vicat Softening Temperature |

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
