# Peer review of "Investigation of the Effects of Multi-Wall and Single-Wall Carbon Nanotubes Concentration on the Properties of ABS Nanocomposites"

_carbon_

Round 1
Reviewer 1 Report
Please see the attached file

Author Response
Reviewer 1
This paper reports the preparation of the physical and electric properties of the ABS/MWCNT and ABS/SWCNT nanocomposites. The thermal, morphology, physical, mechanical, and electric properties of the ABS/MWCNT and ABS/SWCNT nanocomposites were investigated. The paper is suitable for publication in the journal “J. Carbon. Res.” subject to some minor revisions. Following are comments in detail:
1. The full name of “MFI” should be addressed in the abstract.
R1 The melt flow index (MFI) has been modified.
2. The hardness of ABS was not changed or enhanced by the blending of MWCNT and SWCNT. This issue should be discussed in the article.
R2 “This experimental evidence could be explained by the dominant rigidity of the polymer over CNT at ambient temperature.” is added (line 244-245).
3. The number of Tables in article should be newly corrected.
R3 All tables in the manuscript have been modified and re-numbered correctly, in text as well.
4. The sentences in lines 316-323 are not correct. The errors in these sentences should be newly corrected.
R4 Errors in these sentences is due to Cross-reference in Microsoft word. New paragraph has been modified from line 299-313.
5. On page 11, the thermal properties of pure ABS sample should be included in In Table 6.
R5 TGA thermogram and data of pure ABS-filament have been presented in Figure 5 and added in Table 5 (formerly Table 6) and commented.
6. In lines 323 and 324, the authors reported that “the onset temperature (Tonset) and maximum degradation temperature (Td,max) seems not be affected by CNT content.”. In fact, the onset temperature (Tonset) and maximum degradation temperature (Td,max) of ABS/MWCNT nanocomposites were decreased with increasing MWCNT content.
R6 Authors agree with the reviewer and thanks for the observation. Next sentence (line 333-334) has been modified with the comparisons of pure ABS and ABS/CNT nanocomposites.
7. In Table 6, the residue values have not been discussed in the article.
R7 The discussion of residue values was discussed in line 331-332.
8. The MFI values of the ABS/MWCNT and ABS/SWCNT nanocomposites were decreased significantly with increasing MWCNT or SWCNT content. This issue should be further discussed in the article.
R8 The comment of MFI values was added in line 357-358. See also other comments in paragraph 3.3.
9. The S-S curve of pure ABS sample should be included in Figure 7 and Table 4 (on page 12).
R8. Curves and values of ABS-Filament were added in Figure 7 and in Table 6 (page 12) and commented.
Reviewer 2 Report
The paper presents an investigation of the effect of single wall and multi wall carbon nanotube concentration on properties of the polymer based nanocomposite. The subject under study is very relevant since this type of composites has a vast area of applications in various technologies.
Authors have revealed that the increase of nanotubes concentration (for single wall case) results in the monotonous decrease of the elongation to failure and moderate growth of the tensile stress. For the composite with multiwall nanotubes the concentration increase properties modification was revealed to be non-monotonous. The effect is very interesting and deserves a detailed analysis.
Authors have shown that the tensile strength is higher for single wall nanotube nanocomposites. One of probable reasons could be the fact that , for instance the moderate twist deformation can increase the bending stiffness of carbon nanomaterials with reduced dimension [AV Savin, et al. 2019 Mechanics of Materials 137, 103123] (and can be easier realized in single wall carbon nanotubes that in multiwall ones). May be authors could provide some more possible reasons for the enhanced properties of the single wall nanotubes composite. One could also consider the 2-phase effect of 2D carbon nanomaterials stretching that was observed in both graphene and carbon nanotubes which also could contribute to the stretching of the nanotube containing composite [I Evazzade, et al, 2018 Nanotechnology 29 (21), 215704, I Evazzade, 2020 Journal of Micromechanics and Molecular Physics 5 (01), 2050001]
Another thing to be mentioned is the choice of the nanotube concentration. Some reasons for setting concentration values at 5, 7,5 and 10 % should be provided.
After considering the listed points, the paper can be published in the Journal.
Author Response
2.1 Authors have shown that the tensile strength is higher for single wall nanotube nanocomposites. One of probable reasons could be the fact that , for instance the moderate twist deformation can increase the bending stiffness of carbon nanomaterials with reduced dimension [AV Savin, et al. 2019 Mechanics of Materials 137, 103123] (and can be easier realized in single wall carbon nanotubes that in multiwall ones). May be authors could provide some more possible reasons for the enhanced properties of the single wall nanotubes composite. One could also consider the 2-phase effect of 2D carbon nanomaterials stretching that was observed in both graphene and carbon nanotubes which also could contribute to the stretching of the nanotube containing composite [I Evazzade, et al, 2018 Nanotechnology 29 (21), 215704, I Evazzade, 2020 Journal of Micromechanics and Molecular Physics 5 (01), 2050001]
R2.1 Authors appreciate the comments of Reviewer 2. Consequently the discussion has been modified and enriched (see lines 387-394) and two of the suggested references have been added.
2.2 Another thing to be mentioned is the choice of the nanotube concentration. Some reasons for setting concentration values at 5, 7,5 and 10 % should be provided.
R2.2 The values of 5, 7.5 and 10 wt% were selected according to a previous work made by the author’s research group (ref 16). And we also considered that the electrical percolation threshold of nanocomposites was achieved at 0.9 wt % for MWCNT. We intend to achieve highly conductive nanocomposites suitable for thermoelectric applications starting from CNT concentration higher than 4wt%. New sentences have been added in Introduction (line 94-99) with the motivation of the choice.
Reviewer 3 Report
It is interesting article, but it needs some improvements. I suggest to take into account the following comments
The Introduction content includes group citations of articles. More details on the quoted papers should be provided. (i.e. 1-5 or 7-19).
The Introduction also lacks information on laser surface modification of structures with nanotubes, such as:
Polymers 2018, 10 (10), 1091; https://doi.org/10.3390/polym10101091
or Lasers in Manufacturing and Materials Processing 5 (2): 1-14
DOI: 10.1007 / s40516-018-0060-8
or https://doi.org/10.1080/10426914.2015.1019097
lines 187, 197, 230, 286 there are insertions from another language indicating an error
Figs 3 and 4 are insufficiently described in the text. It is worth showing the most important information in the pictures.
316-323 it looks like there is some missing information.
Author Response
Reviewer 3
It is interesting article, but it needs some improvements. I suggest to take into account the following comments
3.1The Introduction content includes group citations of articles. More details on the quoted papers should be provided. (i.e. 1-5 or 7-19).
R3.1 the authors thanks Reviewer 3 for this useful comment. The sequence of references has been modified, and various references have been properly separated, as detailed in the followings. Reference [1-5], ref [1] and [4] of the previous version of the manuscript have been removed. New ref [1,2] have been moved in the first sentence, related to the general information on CNT. Then for each application a single reference has been associated and re-numbered from [3] to [6].
Refs [7-16] (not [7-19]), [9], [10], [11], [12], [13], [16] of the previous version of manuscript have been removed. Only some references (new ref [9-12]) remained, as review papers. The details of removed references ref [10-13] is described in two following paragraphs. Finally, following also the comments of other Reviewers, five others references have been added.
3.2 The Introduction also lacks information on laser surface modification of structures with nanotubes, such as:
Polymers 2018, 10 (10), 1091; https://doi.org/10.3390/polym10101091
or Lasers in Manufacturing and Materials Processing 5 (2): 1-14
DOI: 10.1007 / s40516-018-0060-8
or https://doi.org/10.1080/10426914.2015.1019097
R3.2 A new paragraph about surface treatment of nanofillers is added (line 60- 76).
However, the authors do not consider the three suggested references, because they are beyond the purpose of this manuscript. In fact, the laser modification by surface etching and the ablation processes on the compounded composites are related to different types of composites and for specific purposes; in particular, composites containing Carbon Fiber in view of adhesive bonding, or crosslinked matrix (rubber) in view of improving the hydrophobicity of the rubber, whereas in our case we have Carbon Nanotube and a thermoplastic matrix.
3.3 lines 187, 197, 230, 286 there are insertions from another language indicating an error
R3.3 Errors in these sentences is due to Cross-reference in Microsoft word. All those errors have been removed.
3.4 Figs 3 and 4 are insufficiently described in the text. It is worth showing the most important information in the pictures.
R3.4 Text in 3.2.1. Morphology and density have been added, and also the diameter of filaments.
3.5 lines 316-323 it looks like there is some missing information.
R3.5 Errors in these sentences is due to Cross-reference in Microsoft word. New paragraph has been modified ( line 298-310).
Round 2
Reviewer 3 Report
In my opinion, all needed information has been added to the paper and now it is ready to be published